# [^68^Ga]Ga-PSMA and [^68^Ga]Ga-RM2 PET/MRI vs. Histopathological Images in Prostate Cancer: A New Workflow for Spatial Co-Registration

**DOI:** 10.3390/bioengineering10080953

**Published:** 2023-08-11

**Authors:** Samuele Ghezzo, Ilaria Neri, Paola Mapelli, Annarita Savi, Ana Maria Samanes Gajate, Giorgio Brembilla, Carolina Bezzi, Beatrice Maghini, Tommaso Villa, Alberto Briganti, Francesco Montorsi, Francesco De Cobelli, Massimo Freschi, Arturo Chiti, Maria Picchio, Paola Scifo

**Affiliations:** 1Faculty of Medicine and Surgery, Vita-Salute San Raffaele University, Via Olgettina 58, 20132 Milan, Italy; ghezzo.samuele@hsr.it (S.G.); neri.ilaria@hsr.it (I.N.); mapelli.paola@hsr.it (P.M.); brembilla.giorgio@hsr.it (G.B.); bezzi.carolina@hsr.it (C.B.); t.villa@studenti.unisr.it (T.V.); briganti.alberto@hsr.it (A.B.); montorsi.francesco@hsr.it (F.M.); decobelli.francesco@hsr.it (F.D.C.); chiti.arturo@hsr.it (A.C.); picchio.maria@hsr.it (M.P.); 2Department of Nuclear Medicine, IRCCS San Raffaele Scientific Institute, Via Olgettina 60, 20132 Milan, Italy; savi.annarita@hsr.it (A.S.); samanesgajate.anamaria@hsr.it (A.M.S.G.); 3Department of Radiology, IRCCS San Raffaele Scientific Institute, Via Olgettina 60, 20132 Milan, Italy; 4Department of Pathology, IRCCS San Raffaele Scientific Institute, Via Olgettina 60, 20132 Milan, Italy; maghini.beatrice@hsr.it (B.M.); freschi.massimo@hsr.it (M.F.); 5Department of Urology, Division of Experimental Oncology, Urological Research Institute, IRCCS San Raffaele Scientific Institute, Via Olgettina 60, 20132 Milan, Italy

**Keywords:** PET/MRI, histopathology, co-registration, prostate

## Abstract

This study proposed a new workflow for co-registering prostate PET images from a dual-tracer PET/MRI study with histopathological images of resected prostate specimens. The method aims to establish an accurate correspondence between PET/MRI findings and histology, facilitating a deeper understanding of PET tracer distribution and enabling advanced analyses like radiomics. To achieve this, images derived by three patients who underwent both [^68^Ga]Ga-PSMA and [^68^Ga]Ga-RM2 PET/MRI before radical prostatectomy were selected. After surgery, in the resected fresh specimens, fiducial markers visible on both histology and MR images were inserted. An ex vivo MRI of the prostate served as an intermediate step for co-registration between histological specimens and in vivo MRI examinations. The co-registration workflow involved five steps, ensuring alignment between histopathological images and PET/MRI data. The target registration error (TRE) was calculated to assess the precision of the co-registration. Furthermore, the DICE score was computed between the dominant intraprostatic tumor lesions delineated by the pathologist and the nuclear medicine physician. The TRE for the co-registration of histopathology and in vivo images was 1.59 mm, while the DICE score related to the site of increased intraprostatic uptake on [^68^Ga]Ga-PSMA and [^68^Ga]Ga-RM2 PET images was 0.54 and 0.75, respectively. This work shows an accurate co-registration method for histopathological and in vivo PET/MRI prostate examinations that allows the quantitative assessment of dual-tracer PET/MRI diagnostic accuracy at a millimetric scale. This approach may unveil radiotracer uptake mechanisms and identify new PET/MRI biomarkers, thus establishing the basis for precision medicine and future analyses, such as radiomics.

## 1. Introduction

Prostate cancer (PCa) is the second most common cancer among men worldwide [1]. The accurate detection and delineation of intraprostatic tumor burden are important for diagnosis and treatment planning for patients with primary PCa. In this setting, multi-parametric Magnetic Resonance Imaging (mpMRI) is currently the most comprehensive work-up for non-invasive primary tumor staging. However, accumulating evidence has shown the utility of Positron Emission Tomography (PET), with prostate-specific membrane antigen (PSMA), for the characterization of primary PCa, reporting a higher sensitivity of [^68^Ga]Ga-PSMA-11 PET as compared to mpMRI for tumor localization [2,3,4,5,6,7].

Several studies have examined the diagnostic accuracy of PET with various radiotracers in primary PCa [5,8,9,10]. However, only a limited number of investigations have focused on the correlation between radiotracer uptake and histopathological findings at a detailed level, going beyond the qualitative interpretation of experts.

By quantitatively comparing PET findings with histopathology, it is possible to characterize prostate cancer at a localized level, which is relevant considering that PCa is often heterogeneous and multifocal.

The assessment of the accuracy of PET by means of pathological gold standards requires perfect co-registration between these modalities. The co-registration process could be challenging due to the different resolutions that characterize the imaging modalities and the difficulty in the localization of homologous landmarks, as for PET and pathological images. Moreover, the resection from the body and the formalin fixation induces shrinkage and modifications in the shape of the specimen. Non-rigid registration, in this case, is fundamental to correct for image deformations and to achieve a good matching among different imaging modalities.

Several approaches have been implemented to validate the in vivo images with histological slices [11,12,13,14,15,16,17,18,19,20,21,22,23,24]. Some authors suggested the use of the acquisition of the MRI or Computed Tomography (CT) ex vivo specimen as an intermediate step [11,15,16,17,18,19,23,25,26] since the histopathological specimen is acquired with the same contrast of the in vivo image, while maintaining the deformation due to resection of the histopathology images.

For PET, the use of hybrid systems is fundamental for the intrinsic co-registration with anatomical images that can also be used for the ex vivo acquisition; thus, the use of a fully hybrid PET/MRI system is crucial.

Our study aims to present a co-registration workflow of in vivo PET/MRI prostate cancer examinations acquired using two different tracers ([^68^Ga]Ga-PSMA and [^68^Ga]Ga-RM2), and histopathological digital images. To our knowledge, this is the first paper focusing on a complete co-registration procedure of histology with double-tracer PET/MRI prostate data.

To test our approach, we applied this workflow to the studies of three patients and calculated the target registration error (TRE) of the different steps of the co-registration procedure. Eventually, we compared the co-registered regions of interest (ROIs) of in vivo PET examinations with histopathology tumor ROIs.

## 2. Materials and Methods

### 2.1. Patients

We used images from three patients with biopsy-proven PCa enrolled from 25 January to 29 March 2021, at the IRCCS San Raffaele Scientific Institute, Milan, Italy. Inclusion criteria were age greater than 18 years old at the time of PET/MRI scan, and biopsy-proven high-risk PCa (defined as PSA > 20 ng/mL and/or clinical stage ≥ cT2c and/or biopsy ISUP grade ≥ 4, according to European Association of Urology guidelines [27]) candidate to prostatectomy and pelvic lymphadenectomy. Exclusion criteria were inability to complete the required imaging examinations (i.e., severe claustrophobia), medical condition possibly interfering and significantly affecting study compliance, contraindications to undergo MRI scan (i.e., metallic/conductive or electrically/magnetically active implants without MR-safe or MR-conditional labeling), and evidence of metastatic disease on conventional imaging contraindicating the surgical procedure.

All patients underwent [^68^Ga]Ga-PSMA PET/MRI and, 48 h later, [^68^Ga]Ga-RM2 PET/MRI for staging purposes before radical prostatectomy. This study (EudraCT: 2018-001034-18) was approved by the Institutional Ethics Committee of IRCCS San Raffaele on 4 March 2020, and all patients gave written informed consent to participate in the study.

### 2.2. In Vivo PET/MRI Acquisitions

The two consecutive examinations, [^68^Ga]Ga-PSMA PET/MRI and [^68^Ga]Ga-RM2 PET/MRI were acquired using a PET/MRI system (SIGNA PET/MR 3T, General Electric Healthcare, Waukesha, WI, USA). The complete acquisition protocols have been described elsewhere [28].

The in vivo images that were acquired in the protocols and that were used in this work include (1) the high-statistics pelvis [^68^Ga]Ga-PSMA PET, (2) the high-statistics pelvis [^68^Ga]Ga-RM2 PET, (3) the corresponding MR-based attenuation correction (MRAC) images simultaneously acquired during the two PET/MRI studies, and (4) the 3D T2-weighted images of the prostate, acquired during the high-statistics pelvis [^68^Ga]Ga-RM2 PET/MRI.

### 2.3. Radical Prostatectomy

All patients underwent radical prostatectomy plus extended pelvic lymph node dissection (ePLND) using a robotic-assisted (RARP) approach. Extended lymph node dissection was performed considering that the pre-surgical risk of lymph node invasion was >5%, and it included the removal of fibrofatty tissue along the external iliac vein, with the lateral limit being the genitofemoral nerve and the distal limit being the deep circumflex vein. Proximally, the cranial limit was represented by the crossing between the ureter and common iliac vessels. All fibrofatty tissue within the obturator fossa was removed. The dissection was performed laterally to medially up to the umbilical artery, and the bladder wall represented the medial limit. Lymph nodes along as well as medially and laterally to the internal iliac vessels were also removed.

### 2.4. Preparation of the Ex Vivo Prostate

Immediately after resection, radical prostatectomy’s fresh specimen was received from the operating room; weight and dimensions were recorded. An experienced pathologist removed the seminal vesicles and covered the surface of the specimen with India ink (Germany 1928, Rotring). Afterward, the specimens were marked with strand-shaped fiducial markers visible both on histology and MR images. These markers consisted of cotton thread infused with a 1:40 solution of gadobutrol (Gadovist, Bayer Schering Pharma, Leverkusen, Germany) and India ink. Fiducials were inserted to maximize the difference in the configurations of points for the possible cutting planes. After the insertion of the markers, the specimen was fixed in 10% buffered formalin for about 24 h. The procedure is similar to that of Gibson et al. [29].

### 2.5. Ex Vivo PET/MRI Acquisitions

The resected and formalin fixed prostates were carefully located on ad hoc home-made mold (one for each specimen) with a visible ruler perpendicular to the gland’s long axis (apical–basal), on the axial MR orientation to facilitate the subsequent cut.

An ex vivo MR acquisition of the mold with the fixed prostate specimen was then collected on the same PET/MR system used for in vivo examination but using a 32 channel coil. The mold was carefully positioned in the coil to have the prostate in an axial orientation as defined by the ruler on the mold. The MR acquisition included the following pulse sequences: 2 three-plane localizer 2D SSFSE sequences, a 3D T1-weighted high-resolution sequence (BRAVO, TR = 13.8 ms, TE = 6.3 ms, TI = 450 ms, voxel size: 0.3 × 0.3 × 0.3 mm^3^, 3 NEX), and a 3D T2-weighted high-resolution sequence (CUBE, TR = 2502 ms, TE = 60 ms, voxel size: 0.3 × 0.3 × 0.3 mm^3^, 4 NEX). The total acquisition time was about 50 min.

### 2.6. Specimen Cut and Digitalization

After ex vivo MR acquisition, each specimen positioned on the ad hoc mold was sectioned. The apical and basal parts of the prostate were removed by a transversal cut at 3 mm from the distal and proximal margins, respectively, and then sectioned into slices at 3 mm intervals perpendicularly to the inked surface. The prostate body was step-sectioned at 3 mm intervals, oriented as defined by the ruler, perpendicular to the gland’s long axis (apical–basal). The cut specimen was post-fixed for 24 h in formalin and then dehydrated in graded alcohols, cleared in xylene, embedded in paraffin, and examined histologically as 4 μm-thick whole-mount hematoxylin and eosin-stained sections. For the body of the prostate, sections were used with special molding and inclusion cassettes. The 2014 ISUP/WHO-modified Gleason score and Grade Group were provided [30]. The index lesion was considered as the cancer focus with the highest Gleason score, or the largest focus as measured by the volume in the case of more than one lesion with the same Gleason score. The index lesion site, pT, grading group, extra prostatic extension (yes/no), positive surgical margins (yes/no), and metastatic lymph nodes (number, maximum diameter, and site) were gathered.

The 2D sections were then digitalized and pathological ROIs were annotated by using 3DHISTECH system (Budapest, Hungary).

### 2.7. PET/MR Image Analysis

[^68^Ga]Ga-PSMA and [^68^Ga]Ga-RM2 PET image read-outs were performed on the Advantage Workstation (AW, General Electric Healthcare, Waukesha, WI, USA). The high-statistic PET acquisition bed on the pelvic region and total-body PET examination of both [^68^GA]Ga-PSMA and [^68^Ga]Ga-RM2 PET images were qualitatively interpreted by two experienced nuclear medicine physicians, with knowledge of all the patients’ clinical and imaging information.

For primary tumor assessment, high statistics and total-body PET images were qualitatively evaluated for both [^68^Ga]Ga-PSMA and [^68^Ga]Ga-RM2 PET. The presence of an increased uptake of [^68^Ga]Ga-PSMA and [^68^Ga]Ga-RM2 PET was considered positive for malignancy, and the dominant intraprostatic lesion identified on [^68^Ga]Ga-PSMA and [^68^Ga]Ga-RM2 PET was manually drawn on PET images by an expert nuclear medicine physician slice-by-slice using 3D Slicer software (revision 29402) [31].

### 2.8. Co-Registration of In Vivo, Ex Vivo, and Digitalized Histological Images

The whole procedure is depicted in Figure 1 and consists of 5 different steps:The co-registration of histopathological images to ex vivo 3D T1w MR images;The co-registration of the in vivo 3D T2w MR images to the ex vivo T2w images (originally acquired as the ex vivo T1w images, so naturally co-registered to those images);The application of the calculated transformations to the RM2 PET images;The co-registration of the Water-MRAC images of PSMA PET to the MRAC of the RM2 PET;The application of the calculated transformations to the PSMA PET images.

The histopathological image and its annotation were resized to a matrix of 512 × 512 pixels and then saved as Tag Image File Format (.tiff) files using ImageJ [32] to be loaded on 3D Slicer, where the co-registration was implemented.

First, the histological image and its annotation were set with the same origin of the ex vivo image. After finding the ex vivo slice corresponding to the histological specimen, a semi-automatic landmark registration (“Landmark Registration” tool) was performed using affine transformation (rigid + uniform scale) and a linear interpolator, setting the T1w ex vivo image as “fixed” (reference) and the histopathological image as “moving”. The landmarks were placed on the holes generated by the strand-shaped fiducials visible on both histopathological and MR images (Figure 2A–C). The same transformation matrix was applied to the image of the ROI defining the lesion segmentation. Finally, a volume including the co-registered histopathological image was created with the same dimension, origin, and voxel size of the ex vivo image (Figure 1A).

Afterward, an automatic registration was performed to the T2w in vivo image to align it with the T2w ex vivo one. Specifically, a non-rigid transformation (rigid + BSpline) was applied to the in vivo volume, using mutual information as the similarity metric. Where necessary, a semi-automatic registration (affine) was performed. The two transformation matrices were then applied to the [^68^Ga]Ga-RM2 PET images that are intrinsically co-registered to the 3D T2 in vivo images, since they were acquired simultaneously. The final PET volume was reconstructed using a linear interpolator (Figure 1B).

As the last step, to co-register the two PET images, the transform matrices calculated to align the Water MRAC image acquired during [^68^Ga]Ga-PSMA PET (PSMA MRAC) with the Water MRAC image acquired during [^68^Ga]Ga-RM2 PET (RM2 MRAC) were applied to the [^68^Ga]Ga-PSMA PET volume. Specifically, automatic deformable registration (rigid + BSpline) followed by semi-automatic registration (affine) were performed (Figure 1C). An example of the result obtained from the co-registration procedure is shown in Figure 2D–F.

The same transform matrices were applied to the PET lesion segmentations.

### 2.9. Metrics

The target registration error (TRE) was used as a metric to validate the co-registration procedure and was calculated as the Euclidean distance for each pair of target points, as follows:(1)$$\left\|T(P_{1})-P_{0}\right\|$$
where *P*_0_ is the point chosen on the target slice and *T*(*P*_1_) is the corresponding point chosen on the moved slice to which the transformation matrix (*T*) was applied. Specifically, 8 points were annotated for each slice in all images. An exemplar illustration is shown in Figure 3. The TRE for each pair of sections, namely, (1) histopathology and ex vivo MRI, (2) ex vivo MRI and in vivo MRI, and (3) histopathology and in vivo MRI, is reported as the median and interquartile range (IQR). The Kruskal–Wallis test was used to compare differences in TRE across all groups. Post hoc pair-wise group comparisons were then performed by means of the Wilcoxon signed rank test corrected for multiple testing by the false discovery rate. Corrected *p* values < 0.05 were considered statistically significant.

DICE score was computed to evaluate the diagnostic accuracy of PET examinations. This index was calculated between the regions of interest manually segmented in correspondence of the intraprostatic findings referable to the site of the primary tumor on [^68^Ga]Ga-PSMA and [^68^Ga]Ga-RM2 PET by the nuclear medicine physician and the 2D dominant intraprostatic lesion defined by the pathologist and co-registered to in vivo images. DICE score computation was performed in Python 3.8, while all statistical analyses were performed in R V4.3.0 [33].

## 3. Results

### 3.1. Patients

Clinical and demographic information of the three patients included in this study is reported in Table 1.

### 3.2. Target Registration Error Analysis

The median TRE for the co-registration of histopathology images and ex vivo MRI was 1.15 mm (IQR: 0.64–1.52 mm), that for the co-registration of ex vivo and in vivo MR images was 1.37 mm (IQR: 0.84–1.95), and that for the co-registration of histopathology images and in vivo MRI was 1.59 mm (IQR: 0.96–2.63 mm). The Kruskal–Wallis test showed a significant difference in the TRE among the investigated groups (*p* = 0.039). Post hoc comparisons revealed that the TRE among histopathology and in vivo MRI was significantly higher than the TRE between histopathology and ex vivo MRI (*p* = 0.002), while the other pair-wise differences were not statistically significant (*p* > 0.05, Figure 4).

### 3.3. Diagnostic Accuracy (DICE)

All the patients showed at least one focal tracer uptake (lesion) in both [^68^Ga]Ga-PSMA and [^68^Ga]Ga-RM2 scans. In two out of three patients, the qualitative evaluation of PET scans concurred with the histopathological examination, identifying the dominant intraprostatic lesion in the left peripheral zone in patient n.1 and in the right basal site in patient n.2. Conversely, in patient n. 3, only [^68^Ga]Ga-RM2 PET matched the histopathological examination, reporting pathological uptake in the right posterolateral area of the prostate, while increased [^68^Ga]Ga-PSMA uptake was reported in the left lobe of the organ.

Therefore, the dominant intraprostatic lesion drawn by the pathologist was identified in all patients at [^68^Ga]Ga-RM2 PET examination, resulting in a median DICE score of 0.75 (range: 0.60–0.81), while the median DICE score between the dominant intraprostatic lesion identified by the pathologist and the site of increased intraprostatic uptake referable to the primary tumor defined by the nuclear medicine physician on [^68^Ga]Ga-PSMA PET images was 0.54 (range: 0–0.75) (see Figure 5 for exemplar images). DICE scores for each co-registered slice of the three investigated patients can be found in Table 2.

## 4. Discussion

In this paper, a workflow for the co-registration of the prostate PET images of a double-tracer PET/MRI study to the histopathological images of the prostate-resected specimen is proposed. This workflow includes multiple steps with the fundamental role also of MR images acquired simultaneously to the PET images.

The use of the PET/MRI system enables the intrinsic co-registration of in vivo PET and MRI so that the PET co-registration becomes easier and accurate both for the alignment to the histopathological images and between the two PET/MRI studies acquired on the same patients with two different tracers.

Moreover, the acquisition of the ex vivo images as an intermediate step for the final co-registration is possible only with MRI that has the double advantage of enabling high-resolution images, with the same deformation of the specimen as the histopathological digital images, and with the same contrast as the in vivo images.

The ex vivo MRI acquisition included two high-resolution sequences (3D T1 and 3d T2 images) in the exact same localizations. The two weighted images enable both high contrast of the Gadolinium (T1) and the same contrast as the in vivo images (T2) to be exhibited. Moreover, the T2 images have more anatomical details that might be compared with histopathological images.

In our study, the co-registration procedure was performed using non-rigid transformations to take into account the shrinkage and shearing that may affect the surgical specimen when resected. Moreover, to better compensatefor these deformations, we added as an intermediate step of our procedure the acquisition of the ex vivo MRI of the specimen. In the literature, there are studies that have introduced this step in their co-registration methods [11,12,17,18,19,23]. However, our workflow differs from these studies since we used the MRI acquired simultaneously to co-register the double-tracer PET images to the histologic prostate images.

We co-registered, first, the MR images due to the high anatomical characterization of tissues and then the same transform matrices were applied to PET images acquired simultaneously with MRI. A combination of automatic and semi-automatic registrations was implemented to obtain a perfect matching between images of different imaging modalities. The automatic registration is a straightforward procedure that does not require user interaction and it is useful in a large dataset cohort, but it could not guarantee the good matching. On the other hand, landmark-based registration is an accurate semi-automatic method that is customizable for the specific study [34]. The procedure we used is similar to that of Gibson et al. [18]. The specimen is prepared before the ex vivo MRI acquisition with the thread infused with a solution of gadolinium and India ink to be visible both in MRI and in histopathological images. This step is very important and critical to determine the right slices to co-register and the visible markers are fundamental. However, it requires previous expertise for the identification of the digital histopathology specimen on the corresponding MRI slice. Therefore, combining the two methods was the best compromise.

The first three steps of the procedure can be used for PET/MRI single tracer studies to be co-registered to histopathological images. The last two steps are specific for double-tracer studies to compare the performance of different radiotracers. The use of Water MRAC images is very convenient as the MRI contrast is very strong, the modality is the same, and the co-registration becomes very easy to obtain, while it would be very difficult to co-register two PETs with different tracers that may have very different uptakes.

The accurate co-registration of histopathological specimens and in vivo PET/MR images allows for going beyond the qualitative examination of imaging findings, and objectively determining the accuracy of PET/MRI at the level of single lesions by means of the DICE score. This is especially relevant for patients showing pathological findings at imaging that are subsequently proved as false positives by histopathological assessment of the resected specimen, which occurred in patient n. 3 of this study. Qualitative examination of [^68^Ga]Ga-PSMA PET showed a pathological uptake on the left prostatic lobe, while the pathologist identified the primary tumor on the right postero-lateral side of the organ, thus resulting in a DICE score of zero, as can be seen in Figure 5I.

The target registration error was used to test each step of the co-registration protocol and showed accurate spatial co-registration, resulting in a median TRE of 1.59 mm for the fusion of histopathology and in vivo MRI examinations. Notably, the TRE only minimally increased from the co-registration of histopathology with ex vivo imaging and histopathology with in vivo MR images, thus showing the robustness of the proposed method despite the challenges of non-rigid deformations occurring to the resected prostate. Moreover, our results are in line with those presented in a review [35] that reports a TRE range from a minimum of 0.71 mm to a maximum of 4.9 mm.

One limitation of this study is the low number of patients. The co-registration procedure was tested on five samples, but a larger dataset and more patients are required to better validate the entire procedure and confirm our results.

Possible future developments might involve both the extraction of semi-quantitative parameters from PET images (such as the maximum and mean standardized uptake value) and the application of radiomic analysis [36], potentially improving tumor detection and characterization. This could lead to a better understanding of the biological underpinnings of radiomic features and in unveiling disease-related mechanisms. Notably, a recent work by Chan et al. [37] demonstrated the significance of co-registering histology with PET and MR images, providing accurate ground truth data for predicting tumor detection and grading by means of radiomics.

Furthermore, it would be interesting to implement an automatic co-registration procedure using a deep-learning approach when the number of patients increases. In the literature, there are several studies that investigate this methodology, achieving promising results in terms of time efficiency and comparable performance with respect to conventional image co-registration methods [38,39,40]. This would be helpful when dealing with a large dataset cohort and to improve robustness and reproducibility of the co-registration procedure.

## 5. Conclusions

We present a method to co-register histopathology and dual tracer in vivo PET/MR images for PCa. Our findings demonstrate precise spatial co-registration, enabling objective evaluation of PET/MRI diagnostic accuracy at a millimetric scale. This approach may unravel the pathophysiological mechanisms involved in radiotracer uptake and identify new PET/MR imaging biomarkers, thus laying the groundwork for precision medicine and future analyses, such as radiomic studies.

## Figures and Tables

**Figure 1 bioengineering-10-00953-f001:**
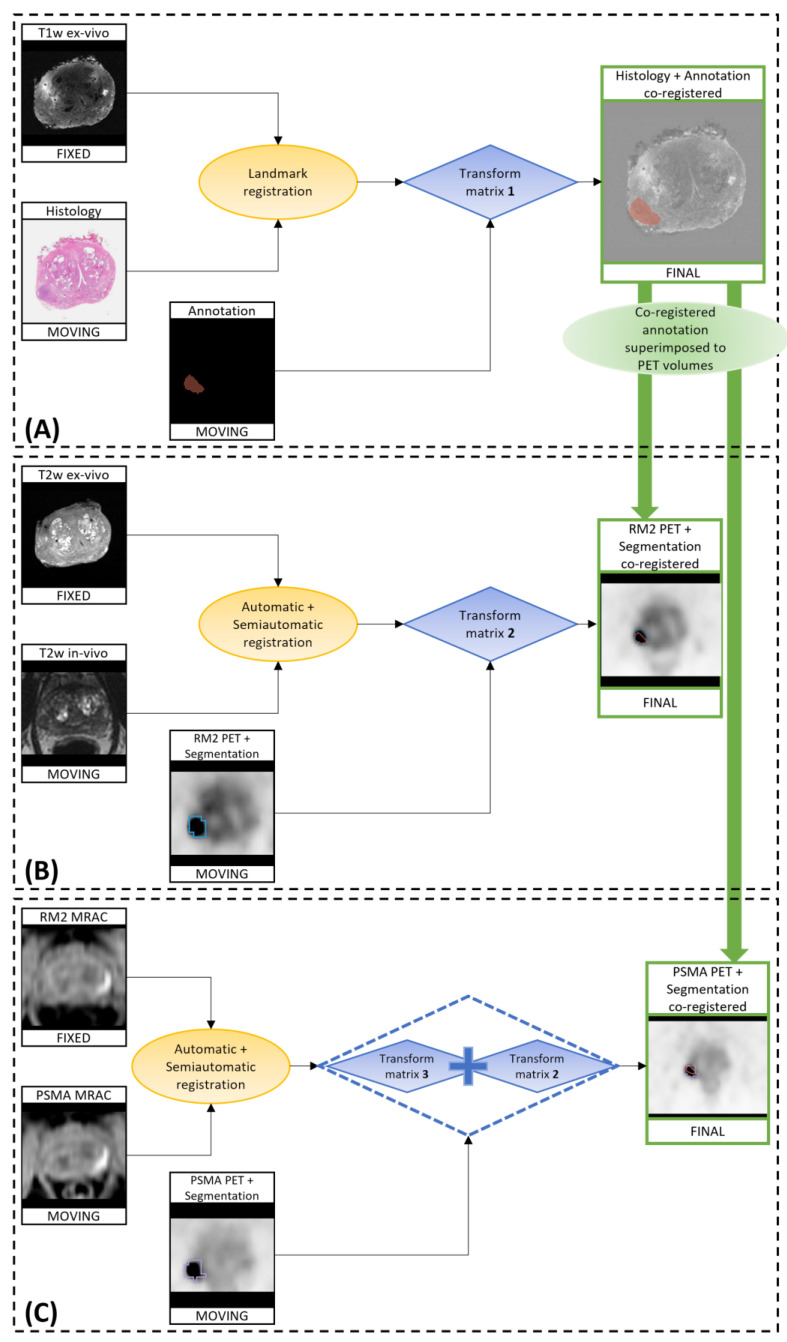
Workflow of the co-registration procedure: (**A**) histological slice co-registered to T1w ex vivo; (**B**) [^68^Ga]Ga-RM2 PET (RM2 PET) co-registered to T2w ex vivo; (**C**) [^68^Ga]Ga-PSMA PET (PSMA PET) co-registered to T2w ex vivo. Final co-registered images are contoured in green. The registration methods are shown inside yellow circles and the transformation matrices are shown inside blue boxes.

**Figure 2 bioengineering-10-00953-f002:**
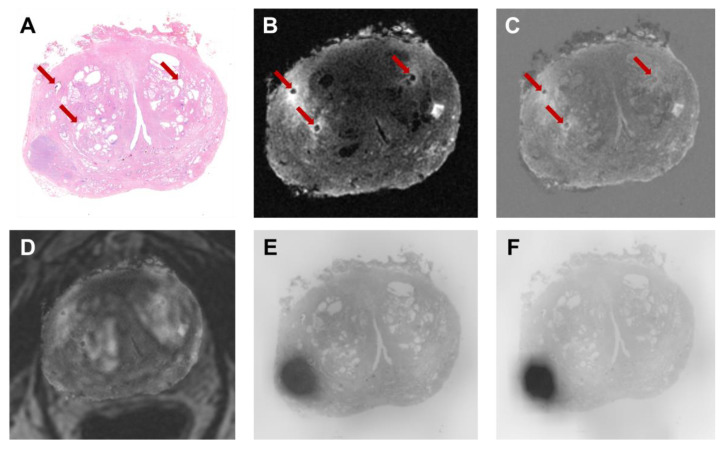
Strand-shaped fiducials (red arrows) visible on (**A**) histopathology, (**B**) T1w MR image, and (**C**) fused histopathology-MR images of Patient 2. An example of results obtained from the co-registration procedure is shown: (**D**) ex vivo MRI overlaid onto T2w in vivo; (**E**) histology overlaid onto [^68^Ga]Ga-RM2 PET; (**F**) histology overlaid onto [^68^Ga]Ga-PSMA PET.

**Figure 3 bioengineering-10-00953-f003:**
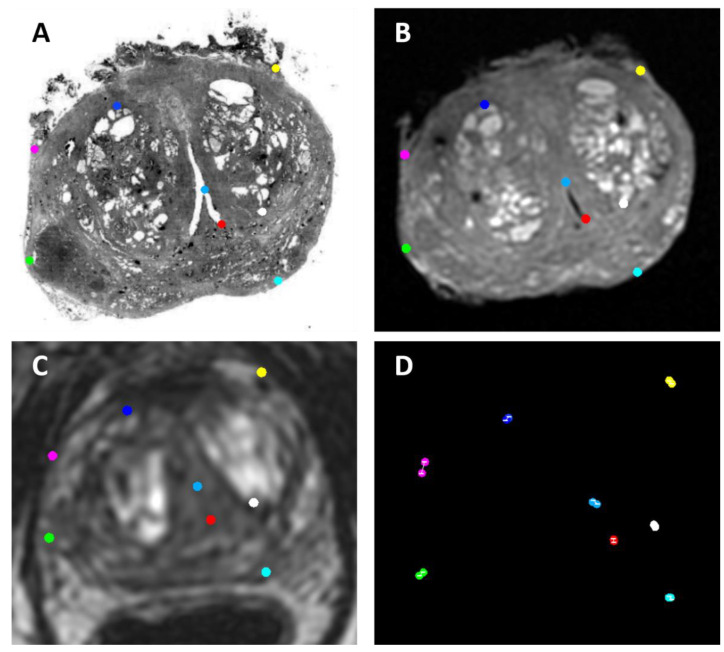
Representative illustration of the 8 color-coded target points defined on (**A**) histological, (**B**) T2w ex vivo, and (**C**) T2w in vivo images. The Euclidean distance between each pair of target points (white line) is shown in (**D**).

**Figure 4 bioengineering-10-00953-f004:**
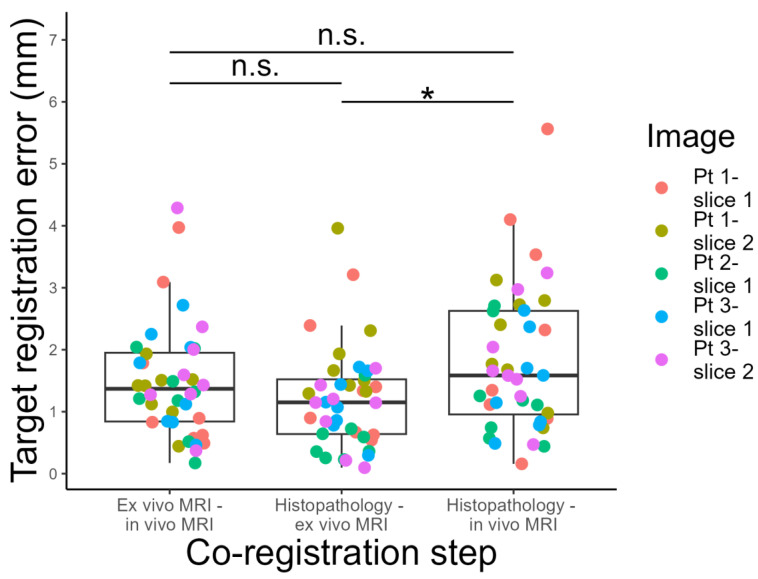
Graphical representation of TRE results for each co-registration step. * adjusted *p* < 0.05; n.s. non significant.

**Figure 5 bioengineering-10-00953-f005:**
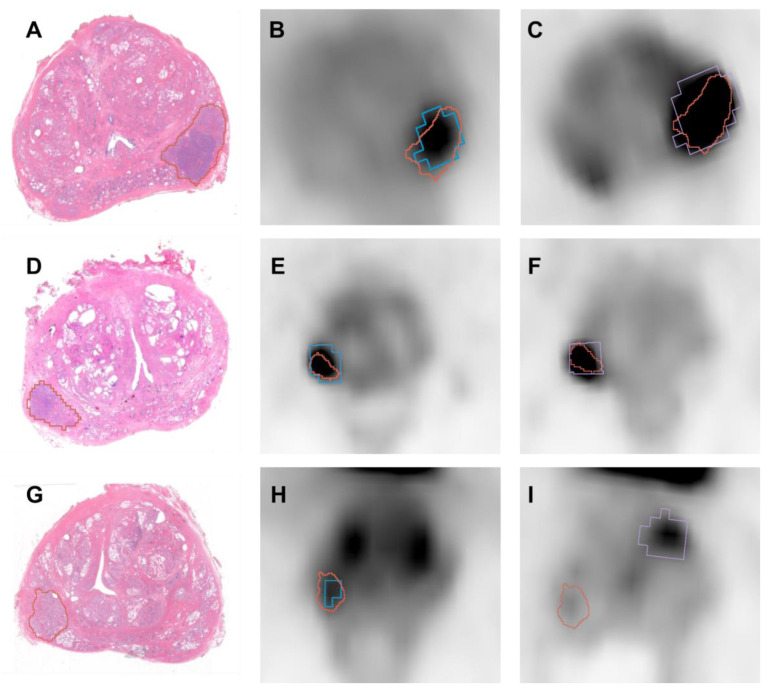
Selected images of two patients (first row: Patient 1; second row: Patient 2; third row: Patient 3): (**A**,**D**,**G**) Histologic slices with annotations of the lesion; (**B**,**E**,**H**) RM2 PET with segmentation (blue) of the tumor uptake and the co-registered histological annotation (red); (**C**,**F**,**I**) PSMA PET with segmentation (violet) of the tumor uptake and the co-registered histological annotation (red).

**Table 1 bioengineering-10-00953-t001:** Demographic and clinical information of the three patients of the study. PSA: prostate-specific antigen; GS: Gleason Score.

N	Age (Years)	PSA Level at Diagnosis (ng/mL)	GS at Biopsy	GS at Prostatectomy	Pathological T Stage
1	74	6.37	9 (5 + 4)	9 (4 + 5)	pT2c
2	64	4.4	8 (4 + 4)	9 (4 + 5)	pT3b
3	66	6.37	9 (4 + 5)	7 (4 + 3)	pT3a

**Table 2 bioengineering-10-00953-t002:** DICE scores of co-registered slices of the three investigated patients.

Patient	[^68^Ga]Ga-RM2 PET-Histology	[^68^GA]Ga-PSMA PET-Histology
Patient 1–slice 1	0.773	0.555
Patient 1–slice 2	0.814	0.542
Patient 2	0.746	0.751
Patient 3–slice 1	0.624	0
Patient 3–slice 2	0.596	0

## Data Availability

The datasets generated during the current study are available from the corresponding author on reasonable request.

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
