# Peer review of "[68Ga]Ga-PSMA and [68Ga]Ga-RM2 PET/MRI vs. Histopathological Images in Prostate Cancer: A New Workflow for Spatial Co-Registration"

_bioengineering, 2023, doi:10.3390/bioengineering10080953_

Round 1

Reviewer 1 Report

In this study, the authors set up a workflow to co-register the prostate PET images with histopathological images using the ex-vivo and in-vivo MRI as an intermediate bridge. Although it seems feasible, more evidences are needed to confirm the accuracy of the co-registration method.

Major Comments:

1.       Besides DICE scores, more metrics can be provided, such as the Target Registration Error.

2.       Did the authors collected multiple sections from the specimen to build the 3D histological images? If yes, why only DICE scores were calculated from only 1~2 slices.

3.       I recommend to add a figure showing the overlay of histopathology, ex vivo MR image, in vivo MR image and PET image.

4.       Please explain why the PSMA PET didn’t match with the histology in Patient 3 in the discussion

5.       Why did the authors use the water-MARC images instead of the respective MR image to co-register the PSMA PET and the RM2 PET.  

Minor comments:

1.       Full name should be given when the abbreviation is used in the first instance, for example, PET, MRI, MRAC, etc.

Author Response

Thank you for the precious comments about our paper. Please see the attachment for our response to the comments.

Reviewer 2 Report

The abstract briefly mentions the proposed workflow for co-registering PET/MRI images with histopathological images but lacks essential information about the novelty and potential impact of the study.

The study mentions only three patients who underwent both [68Ga]Ga-PSMA and [68Ga]Ga-RM2 PET/MRI. Such a small sample size may limit the generalizability of the findings. Consider discussing the limitations of the small sample size and the need for larger studies to validate the results and establish the workflow's reliability.

The manuscript lacks a comparison with existing co-registration methods for prostate PET/MRI and histopathological images. It would be beneficial to include a brief discussion of how the proposed workflow differs from or improves upon existing approaches in the field.

While the paper mentions DICE scores and SUV values as metrics, it does not explain their clinical implications or how they contribute to a deeper understanding of prostate cancer characteristics.

The paper lacks clear concluding remarks summarizing the key findings and their implications. Additionally, it would be valuable to outline potential future directions for research based on the study's outcomes and insights.

The quality of Figure 1 and Figure 2 does not meet the standard required for publication in this journal. The figure's resolution is too low, making it difficult to discern any meaningful information.

English could be improved 

1.   English could be improved – there are some very cumbersome sentences/paragraphs that could be shortened and improved for better readability. In addition, some sentences have words missing so they don’t make grammatical sense. There are also some sentences missing capital letters at the beginning of the sentence.

Author Response

(The authors gave the same response as above.)

Reviewer 3 Report

The comments for revised manuscript based on the initial assessment are the following:

1)      The “To our knowledge, this is the first paper reporting on the co-registration of PET/MRI data with pathology.” could be merged with the last paragraph of the Introduction section. It can be more organized in the writing. Some work published using a similar concept may study the respective authors before this claim. Example: Cancers. 2022 Mar 29;14(7):1727 and European urology. 2016 Nov 1;70(5):829-36.

2)      In Figure 2, the authors may mention the sourcing of histology and MR images. Is it from Patient 2 or 3?

3)      In Figure 3, the authors included all the different images collected from Patient 2 and Patient 3 but did not include Patient 1. As the population of this study is three, the authors can add Patient 1 histology and PET/MRI.

4)      Based on this co-registration of PET/MRI data and Figure 1, can a simulation study be run? It can be a more interesting output for some readers in this field.

5)      The conclusion part needs to be informative. It may include prospects of such co-diagnosis studies and applications.

Author Response

(The authors gave the same response as above.)

Round 2

Reviewer 1 Report

The authors has resopnced to all my concerns. It's sufficiently improved to warrant publication in Bioengineering.

Reviewer 2 Report

Accepted

Accepted

Reviewer 3 Report

The editor may accept the revised manuscript.